# Synthesis and Glycosidase Inhibition of Broussonetine M and Its Analogues

**DOI:** 10.3390/molecules24203712

**Published:** 2019-10-15

**Authors:** Qing-Kun Wu, Kyoko Kinami, Atsushi Kato, Yi-Xian Li, Yue-Mei Jia, George W. J. Fleet, Chu-Yi Yu

**Affiliations:** 1Beijing National Laboratory for Molecular Science (BNLMS), CAS Key Laboratory of Molecular Recognition and Function, Institute of Chemistry, Chinese Academy of Sciences, Beijing 100190, China; lsqkwu@iccas.ac.cn (Q.-K.W.); tamarali@iccas.ac.cn (Y.-X.L.); jiayuemei@iccas.ac.cn (Y.-M.J.); 2University of Chinese Academy of Sciences, Beijing 100049, China; 3Department of Hospital Pharmacy, University of Toyama, 2630 Sugitani, Toyama 930–0194, Japan; kimagure1242@yahoo.co.jp; 4Chemistry Research Laboratory, Department of Chemistry, University of Oxford, Mansfield Road, Oxford OX13TA, UK; 5National Engineering Research Center for Carbohydrate Synthesis, Jiangxi Normal University, Nanchang 330022, China

**Keywords:** broussonetine M, analogue, synthesis, glycosidase inhibition, structure-activity relationship

## Abstract

Cross-metathesis (CM) and Keck asymmetric allylation, which allows access to defined stereochemistry of a remote side chain hydroxyl group, are the key steps in a versatile synthesis of broussonetine M (**3**) from the d-arabinose-derived cyclic nitrone **14**. By a similar strategy, *ent*-broussonetine M (***ent*-3**) and six other stereoisomers have been synthesized, respectively, starting from l-*arabino*-nitrone (***ent*-14**), l-*lyxo*-nitrone (***ent*-3-*epi*-14**), and l-*xylo*-nitrone (**2-*epi*-14**) in five steps, in 26%–31% overall yield. The natural product broussonetine M (**3**) and **10’-*epi*-3** were potent inhibitors of β-glucosidase (IC_50_ = 6.3 μM and 0.8 μM, respectively) and β-galactosidase (IC_50_ = 2.3 μM and 0.2 μM, respectively); while their enantiomers, ***ent*-3** and ***ent*-10’-*epi*-3**, were selective and potent inhibitors of rice α-glucosidase (IC_50_ = 1.2 μM and 1.3 μM, respectively) and rat intestinal maltase (IC_50_ = 0.29 μM and 18 μM, respectively). Both the configuration of the polyhydroxylated pyrrolidine ring and C-10’ hydroxyl on the alkyl side chain affect the specificity and potency of glycosidase inhibition.

## 1. Introduction

In the past several decades, numerous polyhydroxylated pyrrolidine iminosugars have been isolated and synthesized, which has enriched the extended family of iminosugars [1,2,3,4,5,6,7]; these natural products, together with their synthetic analogues, have exhibited a variety of important biological activities and shown great potential as chemotherapeutic agents for an ever widening number of diseases [6,8]. DAB (1,4-dideoxy-1,4-imino-d-arabinitol, **1**) [9] and DMDP (2,5-dihydroxymethyl-3,4-dihydroxypyrrolidine, **2**) [10] are among the most widespread and studied iminosugars (Figure 1). DAB (**1**), isolated in 1985, is a powerful α-glucosidase inhibitor [11,12,13,14,15], whereas DMDP (**2**) is a potent inhibitor of bovine liver β-glucosidase and β-galactosidase (IC_50_ = 9.7 μM and 3.3 μM, respectively) [16]. DAB (**1**) was also found to be a potent inhibitor of glycogen phosphorylase, which made it a potential therapeutic agent for the treatment of diabetes [17]. In contrast, the enantiomers LAB (1,4-dideoxy-1,4-imino-l-arabinitol) (***ent*-1**) [15] and l-DMDP (***ent*-2**) are both potent and specific α-glucosidase inhibitors [16,18].

Many efforts have been devoted to the synthesis and glycosidase inhibition studies of DAB (**1**) [19,20] and various analogues [13,21,22,23,24,25,26,27,28]. Among these studies, the biological evaluation results of α-*C*-1-alkyl-DABs and their l-enantiomers [29] attracted our attention. While α-*C*-1-alkyl-DABs shifted to be β-glucosidase inhibitors with improved potency as the length of the C-1 alkyl chain increased, α-*C*-1-alkyl-LABs upheld similar α-glucosidase inhibitory activities to the parent compound. The above trend in changes of glycosidase inhibition reminded us of a special class of polyhydroxylated pyrrolidine alkaloids, broussonetines, which are a subgroup of natural iminosugars with more than 30 members (Figure 2). Broussonetines were isolated from the branches of the deciduous tree *Broussonetia Kazinoki* SIEB, which is widely distributed in several Asian countries and has been used as a folk medicine [30,31,32,33,34,35,36,37]. Most broussonetines share a common (2*R*, 3*R*, 4*R*, 5*R*)-pyrrolidine moiety with a 13-carbon chain containing various functional groups on the C-5 position. Therefore, they can be structurally regarded as DAB-related iminosugars or α-1-*C*-alkylated-DABs. However, with respect to glycosidase inhibitory activities, broussonetines are better to be considered as analogues of DMDP (**2**), of which most were found to be good β-glucosidase and β-galactosidase inhibitors. For example, broussonetine E (**4**), F (**5**), and G (**6**) show potent β-glycosidase inhibition and have therapeutic potential as antitumor and anti-HIV agents [38]. Broussonetine M (**3**), isolated in 2000 by Kusano’s group, was a potent inhibitor of β-galactosidase from bovine liver (IC_50_ = 8.1 μM) [36].

According to our previous study on the synthesis and glycosidase inhibition of broussonetines [39,40], enantiomers of the above broussonetines would probably demonstrate a similar inhibitory profile by analogy with LAB (***ent*-1**) and l-DMDP (***ent*-2**), and as such [22,26,41,42], these compounds may have potential in the treatment of type II diabetes, cancers, and viral infections [43,44,45,46]. Hence, in this work, broussonetine M (**3**) was selected as the research objective; synthesis and glycosidase inhibition of the natural product and its analogues, including l-enantiomers and pyrrolidine core stereoisomers, were finished, aiming for a better understanding of structure–activity relationship (SAR) of this interesting class of pyrrolidine iminosugars.

Though the first synthesis of broussonetine M (**3**) was accomplished with d-serine as the starting material [47], we have shown that most broussonetines can be efficiently constructed via a general synthetic strategy employing sugar-derived cyclic nitrones [39,40]. The pyrrolidine core of this class of iminosugars can be derived from cyclic sugar nitrones [44,48,49,50] with the corresponding stereochemistry in the hydroxylated pyrrolidine ring, while the various side chains could be installed via cross-metathesis (CM) reactions [46,47,51,52]. This general strategy is capable of synthesizing a number of natural broussonetines, as well as a variety of broussonetine analogues for SAR study; it has been successful in the synthesis of broussonetine I (**7**), J_2_ (**10**), and W (**11**) [39,40]. Therefore, this strategy was applied in the construction of broussonetine M (**3**) and its analogues.

## 2. Results and Discussion

### 2.1. Synthesis of Broussonetine M

Our retrosynthesis for broussonetine M (**3**) is presented in Scheme 1. The precursor **12** of broussonetine M was obtained by the CM reaction between the pyrrolidine **13** and the alcohol **15**. The pyrrolidine **13** was conveniently prepared from d-arabinose-derived cyclic nitrone **14** [53,54,55,56,57]. The alcohol **15**, which contains one stereocenter, was synthesized through asymmetric Keck allylation of aldehyde **16**. In this synthetic route, only the stereocenter in alcohol **15** is constructed by virtue of an asymmetric reaction; of the four stereocenters on the pyrrolidine ring, three were determined by sugar-derived cyclic starting nitrone **14** and the fourth was formed by the high diastereoselectivity of organometallic addition to the nitrone **14**.

For the synthesis of broussonetine M (**3**), addition of Grignard reagent **18**, prepared from 8-bromo-1-octene, to d-*arabino*-nitrone (**14**) afforded the hydroxylamine **19** in high yield and excellent diastereoselectivity; none of the other diastereomers were formed (Scheme 2) [54]. Due to its chemical instability, the hydroxylamine **19** was directly used in the next step of reaction without further purification. Successive zinc reduction and *N*-Cbz protection of the crude hydroxylamine **19** provided the pyrrolidine **13**, as the required CM reaction precursor, in 64% overall yield in three steps. The configuration of the newly formed C-5 chiral center was determined as *R* by nuclear Overhauser effects experiment on **19,** with the observation of the strong correlation between H-5 and H-6a, H-6b (Scheme 2).

The key step in the synthesis of the chiral alkyl alcohol **15** was the asymmetric Keck allylation (Scheme 3) [58]. Treatment of butyl glycol **20** with BnBr/NaH/TBAI in DMF-THF gave the mono-*O*-benzylated alcohol **21** in 91% yield. Swern oxidation of alcohol **21** afforded aldehyde **16** (89%), which on enantioselective Keck allylation using (*S*)-BINOL produced alcohol **15** (93% ee) [59]. Using (*R*)-BINOL as the ligand, alcohol ***ent*-15** was also prepared by the same method (See the Appendix A for HPLC analysis of compounds **15** and ***ent*-15**).

CM reaction between pyrrolidine **13** and alcohol **15** completed the synthesis of broussonetine M (**3**). In this case, the free hydroxyl group of the alcohol **15** was tolerated for the CM reaction with no need for *O*-protection. CM reaction of pyrrolidine **13** and alcohol **15** promoted by Grubbs II catalyst produced olefin **12** as an inseparable *Z/E* mixture in moderate yield (43%). Pd/C-catalyzed hydrogenation of compound **12** in acidic methanol afforded the target product broussonetine M (**3**) in quantitative yield (Scheme 4). Thus, broussonetine M (**3**) was synthesized in five linear steps starting from d-*arabino*-nitrone (**14**) in 28% total yield. The ^1^H- and ^13^C-NMR spectra and the specific rotation of the synthetic broussonetine M (**3**) were all consistent with those reported for the synthetic broussonetine M (**3)** [47], but had some differences with those of natural products [36] (See the Appendix A for comparison of NMR data). Since the structure and configuration of product **3** were ensured by all the synthetic materials and procedures, as in the work of Alberto Marco et al. [47], the NMR spectra differences may be explained by minute pH variation or metal impurities [60,61].

### 2.2. Synthesis of Analogues of Broussonetine M

Glycosidase inhibition by pyrrolidine iminosugars are among those that are most difficult to predict [16]; minor modification of the iminosugar can lead to a distinct change of the inhibition profile. In order to explore the preliminary structure–activity relationship of this type of iminosugar, seven analogues of broussonetine M (**3**), including its l-enantiomer and other six stereoisomers, were then prepared. By the same strategy as that for pyrrolidine **13**, synthesis of ***ent*-13**, ***ent*-3-*epi*-13**, and **2-*epi*-13** were accomplished from the corresponding sugar-derived cyclic nitrones ***ent*-14**, ***ent*-3-*epi*-14**, and **2-*epi*-14**. The configurations of the newly constructed chiral centers in these compounds were all confirmed by NOE experiments. With these pyrrolidines in hand, CM reaction with alcohol **15**, and subsequent hydrogenation, provided the target products, i.e., ***ent*-10’-*epi*-3**, ***ent*-3,10’-di-*epi*-3**, and **2-*epi*-3**, of which, the C10’-hydroxyls retained *S* configuration as that of the natural product (Table 1).

In order to evaluate the influence of the C10’-hydroxyl on glycosidase inhibition, the C10’-epimers of broussonetine M (**3**), including **10’-*epi*-3**, ***ent*-3**, ***ent*-3-*epi*-3**, and **2,10’-di-*epi*-3**, were also synthesized from the above four pyrrolidines and alcohol ***ent*-15** by the same strategy.

### 2.3. Glycosidase Inhibition

The synthetic broussonetine M (**3**) and its analogues were all assayed as potential glycosidase inhibitors of a range of enzymes, as shown in Table 2 and Table 3.

According to the results, all the compounds showed moderate to potent inhibition of β-galactosidase from bovine liver; **10’-*epi*-3** was the most potent inhibitor (IC_50_ = 0.2 μM). The natural product broussonetine M (**3)** and **10’-*epi*-3** showed potent inhibition toward β-glucosidase (IC_50_ = 6.3 μM and 0.8 μM, respectively) and β-galactosidase (IC_50_ = 2.3 μM and 0.2 μM, respectively) from bovine liver, which were consistent with DMDP (**2**) and other DAB or DMDP derivatives with a long alkyl side chain on C-1 position [29]. However, different from DMDP (**2**), compounds **3** and **10’-*epi*-3** lost their inhibitions against α-glucosidase, α-mannosidase, and trehalase, whereas they showed moderate β-glucuronidase inhibition. Furthermore, **10’-*epi*-3** with C-10’ having *R* configuration was about 4 times better as an inhibitor than broussonetine M (**3**) with C-10’ having *S* configuration. 

*Ent*-broussonetine M (***ent*-3)** and ***ent*-10’-*epi*-3** exhibited potent and more selective inhibition of α-glucosidase from rice (IC_50_ = 1.2 μM and 1.3 μM) and rat intestinal maltase (IC_50_ = 0.29 μM and 18 μM), analogous to the inhibition by l-DMDP [16,18], *ent*-broussonetine I, J_2_ [39], and other LAB or l-DMDP derivatives with a long alkyl chain on the C1 position [29,62]. ***Ent*-3-*epi*-3** and ***ent*-3,10-di-*epi*-3** showed moderate and broad inhibition against α-glucosidase, β-glucosidase, β-galactosidase, and α-l-fucosidase, while l-*altro*-DMDP was a selective moderate α-glycosidase inhibitor. d-*gluco*-DMDP exhibited moderate inhibition against α-glucosidase. For d-*gluco*-DMDP-related broussonetine M analogs, compounds **2-*epi*-3** and **2,10-di-*epi*-3** both showed moderate inhibition against β-glucosidase, β-galactosidase, and β-glucuronidase; **2,10’-di-*epi*-3** was a potent inhibitor of α-glucosidase, while **2-*epi*-3** did not show such inhibition. The results indicate that the correct configuration at C-10’ is essential for α-glucosidase inhibition.

Therefore, configurations of both the pyrrolidine ring and C-10’ have significant influences on glycosidase inhibition of broussonetine M and its analogs. In detail, comparison of the inhibitory profiles of *ent*-broussonetine M (***ent*-3)** and ***ent*-10’-*epi*-3** with those of broussonetine M (**3)** and **10’-*epi*-3** showed high similarity to those of DAB and LAB derivatives (or DMDP and l-DMDP derivatives), of which we have previously reported opposite inhibitions toward α- and β-glycosidases for enantiomers [16,18,42]. In contrast, inhibitory activities of other broussonetine analogues with l-*altro*-DMDP and d-*gluco*-DMDP pyrrolidine cores are more difficult to predict, but the presence of the 13-carbon chains basically narrowed down the inhibitory profiles.

The structure–activity relationship uncovered in this work would be helpful in the research and development of new glycosidase inhibitors.

## 3. Materials and Methods

### 3.1. General Methods

NMR spectra was recorded at 300 MHz, 400 MHz, or 500 MHz (^1^H-NMR) and 75 MHz, 100 MHz, or 125 MHz (^13^C-NMR) in CDCl_3_ (with TMS as internal standard), C_5_D_5_N, or CD_3_OD (with solvent signal as internal standard). High-resolution mass spectra (HRMS) were performed on a LTQ/FT linear ion trap mass spectrometer. All reagents were used as received without any further purification or prepared, as described in the literature. CH_2_Cl_2_ was freshly distilled from CaH_2_. Tetrahydrofuran was distilled from sodium benzophenone. TLC plates were visualized by ultraviolet light or by treatment with a spray of Pancaldi reagent ((NH_4_)_6_MoO_4_, Ce(SO_4_)_2_, H_2_SO_4_, H_2_O) or a 0.5% solution of KMnO_4_ in acetone. Column chromatography was performed on a flash column chromatography with silica gel (200–300 mesh). Polarimetry was determined using an Optical Activity AA-10R polarimeter with concentrations (*c*) given in grams per 100 mL. IR data were measured as films on KBr plates and are given only when relevant functions are present. Chiral HPLC analyses were performed on an Agilen 1100 Series using a Daicel Chiralpak (OD-H) column with hexanes/*i*-PrOH as the eluent.

### 3.2. Material and Methods for the Enzyme Inhibition Assay

With rat intestinal maltase as an exception, other enzymes were purchased from Sigma-Aldrich Chemical Co. (St. Louis, Mo. USA). Brush border membranes prepared from rat small intestine according to the method of Kessler et al. [63] were assayed at pH 6.8 for rat intestinal maltase using maltose. The released d-glucose was determined colorimetrically using the Glucose CII-test Wako (Wako Pure Chemical Ind.; Osaka, Japan). Other glycosidase activities were determined using an appropriate *p*-nitrophenyl glycoside as substrate in a buffer solution at the optimal pH value of each enzyme. The reaction was stopped by adding 400 mM Na_2_CO_3_. The released *p*-nitrophenol was measured spectrometrically at 400 nm [16].

### 3.3. Chemistry

#### 3.3.1. Synthesis of 4-(benzyloxy)butan-1-ol (**21**)

NaH (60%, 11.52 g, 0.48 mol) was added slowly to a solution of butane-1,4-diol **20** (36.05 g, 0.4 mol) in dry THF (300 mL) at 0 °C. After stirring at room temperature for 0.5 h, tetrabutylammonium iodide (0.74 g, 8.0 mmol) was added, followed by a dropwise solution of benzyl bromide (68.41 g, 0.4 mol) in THF while the temperature was kept between 45 °C and 50 °C. The reaction mixture was stirred for 2 h and then quenched by aq. NH_4_Cl solution (20 mL). Water (200 mL) was added and the organic layer separated; the aqueous layer was extracted with EtOAc (200 mL × 3). The combined organic phases were dried over MgSO_4_, filtered, and concentrated in vacuo. The crude product was purified by flash column chromatography (silica gel, petroleum ether/EtOAc = 3/1) to give 4-(benzyloxy)butan-1-ol **21** (light yellow oil, 69.9 g, 97% yield). ^1^H-NMR (300 MHz, CDCl_3_) *δ* 7.35–7.28 (m, 1H), 4.52 (s, 2H), 3.68–3.63 (d, *J* = 5.8 Hz, 2H), 3.54–3.50 (d, *J* = 5.9 Hz, 2H), 1.81–1.60 (m, 4H); ^13^C-NMR (75 MHz, CDCl_3_) δ 138.2, 128.4, 127.7, 127.7, 73.1, 70.4, 62.7, 30.1, 26.7.

#### 3.3.2. Synthesis of 4-(benzyloxy)butanal (**16**)

A solution of DMSO (8.74 mL, 0.26 mol) in dry CH_2_Cl_2_ (20 mL) was added dropwise to a solution of (COCl)_2_ (24.57 mL, 0.29 mol) in dry CH_2_Cl_2_ (100 mL) at –78 °C. The mixture was stirred for 5 min. A solution of 4-(benzyloxy)butan-1-ol **21** (43.3 g, 0.24 mol) in dry CH_2_Cl_2_ (50 mL) was then added dropwise while the temperature was kept below −65 °C. After 15 min, NEt_3_ (166.94 mL, 1.2 mol) was added dropwise. After stirring for 10 min at −78 °C, the reaction mixture was allowed to warm to room temperature and diluted with CH_2_Cl_2_ (200 mL). The organic layer was washed with brine (2 × 100 mL). The combined organic extracts were dried over MgSO_4_, filtered, and concentrated under reduced pressure. Purification by flash chromatography on silica gel (petroleum ether/EtOAc = 10/1) afforded 4-(benzyloxy)butanal **16** (39.8 g, 93% yield) as a yellow oil. ^1^H-NMR (300 MHz, CDCl_3_) *δ* 9.77 (t, *J* = 1.6 Hz, 1H), 7.35–7.30 (m, 5H), 4.48 (s, 2H), 3.50 (t, *J* = 5.9 Hz, 2H), 2.54 (dt, *J* = 7.2, 1.6 Hz, 1H), 1.98–1.91 (m, 2H); ^13^C-NMR (75 MHz, CDCl_3_) *δ* 202.4, 138.3, 128.4, 127.7, 73.0, 69.1, 41.0, 22.6.

#### 3.3.3. General Procedure for Synthesis of (*R*)-7-(benzyloxy)hept-1-en-4-ol (**15**) and (*S*)-7-(benzyloxy)hept-1-en-4-ol (**ent-15**), with Alcohol **15** as an Example

Under Ar atmosphere, (*S*)-BINOL (52 mg, 0.2 mmol) and Ti(O*^i^*Pr)_4_ (45 mg, 0.2 mmol) were added to a solution of dried 4 Å molecular sieves (2.2 g) in CH_2_Cl_2_ (20 mL). The reaction mixture was heated at reflux for 1 h, and then allowed to cool to room temperature. A solution of aldehyde **16** (356 mg, 2 mmol) in CH_2_Cl_2_ (15 mL) was added to the reaction mixture. After stirring for 0.5 h at room temperature, the solution was cooled to −78 °C and allyltributyltin (993 mg, 3 mmol) was added dropwise. The reaction mixture was stirred for an additional 20 min at −78 °C, then kept at −20 °C. After 12 h, the reaction mixture was filtered through a pad of celite into a 200 mL flask that contained a stirred sat. aq. NaHCO_3_ solution (50 mL); the resulting reaction mixture was stirred for 1 h. Then, the layers were separated and the aqueous layer was extracted with CH_2_Cl_2_ (3 × 50 mL). The combined organic extracts were dried with MgSO_4_; subsequent removal of all volatiles under reduced pressure and column chromatography of the residue on silica gel (petroleum ether/EtOAc = 20/1) afforded homoallylic alcohol **15** (729.7 mg, 83% yield) as colorless oil. 

Data for **15**: [α]_D_
^20^ +4.14 (*c* 3.85 in CH_2_Cl_2_); HPLC analysis: 92.6% ee [Daicel CHIRALPAK OD-H column, 20 °C, 220 nm, hexane/*i*-PrOH = 95:5, 1 mL/min, 20.8 min (major), 23.9 min (minor)]; *ν*_max_/cm^−1^: 3401 (s), 3068 (w), 2924 (s), 1679 (w), 1453 (m), 1096 (s), 1021 (m), 697 (m); ^1^H-NMR (300 MHz, CDCl_3_) *δ* 7.39–7.23 (m, 5H), 5.93–5.75 (m, 1H), 5.15–5.10 (m, 1H), 5.08 (t, *J* = 1.2 Hz, 1H), 4.50 (s, 2H), 3.64 (tt, *J* = 8.1, 4.5 Hz, 1H), 3.50 (t, *J* = 6.0 Hz, 2H), 2.52 (br, 1H), 2.26–2.15 (m, 2H), 1.77–1.60 (m, 3H), 1.54–1.45 (m, 1H); ^13^C-NMR (75 MHz, CDCl_3_) *δ* 138.3, 135.1, 128.4, 127.7, 127.7, 117.7, 73.0, 70.6, 70.5, 42.0, 34.0, 26.2; HRMS(ESI) calcd for C_14_H_21_O_2_^+^ [M + H]^+^ 243.13555, found 243.13564.

Data for ***ent*-15**: colorless oil; yield: 86%; [α]_D_
^20^ +4.08 (*c* 4.45 in CH_2_Cl_2_); HPLC analysis: 92.8% ee [Daicel CHIRALPAK OD-H column, 20 °C, 220 nm, hexane/*i*-PrOH = 95:5, 1 mL/min, 21.3 min (minor), 23.1 min (major)]; *ν*_max_/cm^−1^: 3401 (s), 3068 (w), 2924 (s), 1679 (w), 1453 (m), 1096 (s), 1021 (m), 697 (m); ^1^H-NMR (400 MHz, CDCl_3_) *δ* 7.38–7.31 (m, 4H), 7.31–7.26 (m, 1H), 5.90–5.76 (m, 1H), 5.15–5.12 (m, 1H), 5.10 (s, 1H), 4.50 (s, 2H), 3.7–3.61 (m, 1H), 3.51 (t, *J* = 6.0 Hz, 2H), 2.36 (d, *J* = 3.2 Hz, 1H), 2.32–2.24 (m, 1H), 2.23–2.14 (m, 1H), 1.82–1.70 (m, 2H), 1.70–1.60 (m, 1H), 1.50 (m, 1H); ^13^C-NMR (100 MHz, CDCl3) *δ* 138.3, 135.1, 128.4, 127.7, 127.7, 117.7, 73.0, 70.6, 70.5, 42.0, 34.0, 26.2; HRMS(ESI) calcd for C_14_H_21_O_2_Na^+^ [M + Na]^+^ 243.13555, found 243.13544.

#### 3.3.4. General Procedure for Synthesis of Compounds **19**, **ent-19**, **ent-3-epi-19**, and **2-epi-19**, with **19** as an Example

Part of the solution of 8-bromo-1-octene (573.3 mg, 3.0 mmol) in THF (2 mL) was quickly added via syringe to a stirred solution of Mg (1.16 g, 5.0 mmol) and I_2_ (cat.) in THF (5 mL) under Ar atmosphere. The mixture was heated until the color disappeared; then, the remaining 8-bromo-1-octene was added dropwise. After the addition was completed, the resulting reaction mixture was heated to reflux for 1 h and then was allowed to cool to room temperature. The prepared Grignard reagent was added slowly to a solution of d-*arabino*-nitrone (**14**) (417.5 mg, 1.0 mmol) in THF (10 mL) via syringe at 0 °C under Ar atmosphere. The reaction mixture was stirred for 0.5 h; then sat. aq. NH_4_Cl was added to quench the reaction. The organic layer was separated and the aqueous layer was extracted with EtOAc (3 × 20 mL). The combined organic phases were dried over MgSO_4_ and filtered; the solvent was removed under reduced pressure to give the crude product hydroxylamine **19**, which was used without further purification because of its instability. The sample for structure characterization was purified by flash column chromatography on silica gel (petroleum ether/EtOAc = 5/1) as a colorless syrup. 

Data for (2*R*,3*R*,4*R*,5*R*)-3,4-bis(benzyloxy)-2- ((benzyloxy)methyl)-1-hydroxyl-5-(oct-7-en-1-yl)pyrrolidine **(19)**: [α]_D_
^20^ -8.6 (*c* 1.2 in CH_2_Cl_2_); *ν*_max_/cm^−1^: 3030 (w), 2926 (s), 2855 (s), 1454 (m), 1362 (w), 1097 (s), 735 (m), 697 (s); ^1^H-NMR (400 MHz, CDCl_3_) *δ* 7.32–7.24 (m, 15H), 5.80 (ddt, *J* = 16.9, 10.2, 6.6 Hz, 1H), 5.01–4.91 (m, 2H), 4.56–4.42 (m, 6H), 3.95–3.92 (m, 1H), 3.80–3.76 (m, 2H), 3.58 (dd, *J* = 9.2, 6.9 Hz, 1H), 3.54–3.50 (m, 1H), 3.17 (dt, *J* = 7.5, 5.4 Hz, 1H), 2.04–1.99 (m, 2H), 1.88–1.83 (m, 1H), 1.50–1.43 (m, 1H), 1.42–1.28 (m, 8H); ^13^C-NMR (100 MHz, CDCl_3_) *δ* 139.3, 138.3, 138.2, 138.2, 128.5, 128.5, 128.1, 128.0, 127.8, 127.8, 127.7, 114.3, 86.8, 84.7, 73.5, 71.8, 71.8, 70.2, 70.1, 68.4, 33.9, 29.8, 29.2, 29.0, 26.7; HRMS(ESI) calcd for C_34_H_44_O_4_N^+^ [M + H]^+^ 530.32649, found 530.32565.

Data for (2*S*,3*S*,4*S*,5*S*)-3,4-bis(benzyloxy)-2-((benzyloxy)methyl)-1-hydroxyl-5-(oct-7-en-1-yl) pyrrolidine **(*ent*-19)**: colorless syrup; [α]_D_
^20^ + 7.8 (*c* 2.8 in CH_2_Cl_2_); *ν*_max_/cm^−1^: 3030 (w), 2926 (s), 2856 (s), 1454 (m), 1362 (w), 1099 (s), 735 (m), 697 (s); ^1^H-NMR (400 MHz, CDCl_3_) *δ* 7.31–7.25 (m, 15H), 6.60 (br, 1H), 5.80 (ddt, *J* = 16.9, 10.2, 6.7 Hz, 1H), 5.00–4.91 (m, 2H), 4.56–4.42 (m, 6H), 3.94 (t, *J* = 3.2 Hz, 1H), 3.80–3.77 (m, 2H), 3.60–3,56 (dd, *J* = 9.2, 6.9 Hz, 1H), 3.54–3.50 (m, 1H), 3.16 (dt, *J* = 7.2, 5.4 Hz, 1H), 2.02 (q, *J* = 6.9 Hz, 2H), 1.89–1.84 (m, 1H), 1.59–1.43 (m, 1H), 1.37–1.22 (m, 8H); ^13^C-NMR (100 MHz, CDCl_3_) *δ* 139.3, 138.3, 138.2, 138.2, 128.5, 128.4, 128.4, 128.1, 128.0, 127.8, 127.8, 127.7, 114.3, 86.8, 84.7, 73.4, 71.8, 71.7, 70.2, 70.0, 68.4, 33.9, 29.8, 29.2, 29.0, 26.6; HRMS(ESI) calcd for C_34_H_44_O_4_N^+^ [M + H]^+^ 530.32649, found 530.32550.

Data for (2*S*,3*R*,4*S*,5*S*)-3,4-bis(benzyloxy)-2-((benzyloxy)methyl)-1-hydroxyl--5-(oct-7-en-1-yl) pyrrolidine (***ent*-3-*epi*-19**)**:** colorless syrup; [α]_D_
^20^ −24.5 (*c* 4.5 in CH_2_Cl_2_); ν_max_/cm^−1^: 3030 (w), 2926 (s), 2855 (s), 1454 (m), 1363 (w), 1098 (br), 734 (m), 697 (s); ^1^H-NMR (400 MHz, CDCl_3_) *δ* 7.30–7.24 (m, 15H), 6.64 (br, 1H), 5.84–5.74 (ddt, *J* = 16.9, 10.2, 6.7 Hz, 1H), 5.03–4.88 (m, 2H), 4.68–4.43 (m, 6H), 4.19 (t, *J* = 5.1 Hz, 1H), 3.86–3,76 (m, 2H), 3.68 (dd, *J* = 6.8, 5.3 Hz, 1H), 3.56–3.51 (m, 1H), 3.29–3.24 (m, 1H), 2.04–1.99 (m, 2H), 1.75–1.69 (m, 1H), 1.49–1.28 (m, 9H); ^13^C-NMR (100 MHz, CDCl_3_) *δ* 139.2, 138.5, 138.4, 138.2, 128.6, 128.4, 128.4, 128.2, 127.9, 127.9, 127.8, 127.8, 127.7, 127.7, 127.0, 114.3, 82.9, 76.8, 73.6, 73.5, 72.7, 70.2, 69.3, 67.4, 33.9, 30.1, 29.8, 29.2, 29.0, 26.9; HRMS(ESI) calcd for C_34_H_44_O_4_N^+^ [M + H]^+^ 530.32649, found 530.32573.

Data for (2*S*,3*R*,4*R*,5*R*)-3,4-bis(benzyloxy)-2-((benzyloxy)methyl)-1-hydroxyl-5-(oct-7-en-1-yl) pyrrolidine (**2-*epi*-19**)**:** colorless syrup; [α]_D_
^20^ + 22.8 (*c* 2.5 in CH_2_Cl_2_); *ν*_max_/cm^−1^: 3290 (br), 3028 (w), 2925 (s), 2856 (s), 1453 (m), 1353 (w), 1112 (s), 729 (s), 693 (s); ^1^H-NMR (400 MHz, CDCl_3_) *δ* 7.36–7.23 (m, 15H), 5.85–5.75 (ddt, *J* = 16.9, 10.2, 6.7 Hz, 1H), 5.32 (s, 1H), 5.01–4.91 (m, 2H), 4.57–4.50 (m, 4H), 4.42–4.32 (m, 2H), 3.94–3.87 (m, 2H), 3,81–3.78 (m, 1H), 3,53–3.52 (m, 1H), 3.36–3.31 (dt, *J* = 8.0, 5.3 Hz, 1H), 2.85–2.80 (dt, *J* = 8.0, 5.3 Hz, 1H), 2.05–2.00 (dd, *J* = 7.1, 6.6 Hz, 1H), 1.81–1.75 (m, 1H), 1.57–1.50 (m, 1H), 1.42–1.28 (m, 8H); ^13^C-NMR (100 MHz, CDCl_3_) *δ* 139.3, 138.3, 138.2, 138.0, 128.5, 128.0, 127.9, 127.9, 127.8, 127.8, 114.3, 85.4, 79.8, 73.6, 73.1, 72.3, 71.6, 69.7, 68.0, 33.9, 32.8, 29.8, 29.2, 29.0, 26.2; HRMS(ESI) calcd for C_34_H_44_O_4_N^+^ [M + H]^+^ 530.32649, found 530.32577.

#### 3.3.5. General Procedure for Synthesis of Compounds **13**, **ent-13**, **3-epi-ent-13**, and **2-epi-13**, with **13** as an Example

Zinc powder (653.8 mg, 10 mmol) was added to a suspension of Cu(OAc)_2_ (18.2 mg, 0.1 mmol) in AcOH (10 mL) and the reaction mixture was stirred for 0.5 h. Then, a solution of the crude hydroxylamine **19** in AcOH (5 mL) was added and the reaction mixture was stirred for 10 h. The solid was removed by filtration and all volatiles were removed under reduced pressure. The residue was dissolved in EtOAc (20 mL) and sat. aq. NaHCO_3_ was added to neutralize the solution. The resulting precipitate was removed by filtration and the organic layers were collected; the aqueous layer was extracted with EtOAc (3 × 10 mL). The combined organic layer was dried over MgSO_4_, filtered, and concentrated under reduced pressure to give the crude amine, which was used in the next step without further purification. NaHCO_3_ (252.0 mg, 3.0 mmol) and CbzCl (255.9 mg, 1.5 mmol) were added slowly to a stirred solution of the crude amine in methanol (10 mL) and the reaction mixture was stirred at room temperature for 6 h. Then sat. aq. NaHCO_3_ (20 mL) was added to quench the reaction and EtOAc (20 mL) was added. The organic layer was separated and the aqueous layer was extracted by EtOAc (3 × 10 mL). The combined organic layers were dried over MgSO_4_, filtered, and the solvent was removed under reduced pressure. Purification by flash chromatography on silica gel (petroleum ether/EtOAc = 15/1) afforded the carbamate **13** as light yellow syrup (423 mg, yield: 64% for 3 steps).

Data for (2*R*, 3*R*, 4*R*, 5*R*)-3,4-bis(benzyloxy)-1-benzyloxycarbonyl-2-((benzyloxy)methyl)-5-(oct-7-en-1-yl)pyrrolidine (**13**): [α]_D_
^20^ −36.4 (*c* 2.25 in CH_2_Cl_2_); *ν*_max_/cm^−1^: 2926 (m), 2855 (w), 1700 (vs), 1407 (m), 1347 (w), 1093 (m), 696 (m); ^1^H-NMR (400 MHz, CDCl_3_) *δ* 7.38–7.23 (m, 17H), 7.23–7.15 (m, 3H), 5.86–5.72 (m, 1H), 5.22–5.15 (m, 1H), 5.05 (s, 0.5H), 5.02–5.01 (m, 1H), 4.96–4.92 (m, 1.5H), 4.65–4.57 (m, 1.5H), 4.47–4.32 (m, 4.5H), 4.27–4.23 (m, 0.5H),4.15–4.12 (m, 1.5H), 4.07–4.03 (m, 0.5H), 3.86–3.71 (m, 2.5H), 3.50–3.44 (m, 1H), 2.09–1.94 (m, 2.5H), 1.78– 1.48 (m, 1.5H), 1.29 (m, 4.5H), 1.16 (m, 3.5H); ^13^C-NMR (100 MHz, CDCl_3_) *δ* 153.6, 153.2, 138.1, 138.1, 137.5, 137.3, 136.9, 136.9, 136.7, 136.6, 135.6, 135.6, 127.5, 127.4, 127.3, 127.3, 127.1, 126.99, 126.95, 126.69, 126.65, 126.60, 126.57, 126.52, 126.46, 113.2, 113.1, 83.4, 82.3, 82.1, 81.0, 72.0, 71.9, 70.1, 70.0, 69.8, 67.7, 66.8, 65.8, 65.7, 64.0, 63.6, 61.8, 61.5, 32.7, 32.7, 30.4, 29.1, 28.2, 28.0, 27.8, 27.8, 27.7, 25.5, 25.5; HRMS(ESI) calcd for C_42_H_49_O_5_NNa^+^ [M + Na]^+^ 670.35029, found 670.34943.

Data for (2*S*,3*S*,4*S*,5*S*)-3,4-bis(benzyloxy)-1-benzyloxycarbonyl-2-((benzyloxy)methyl)-5-(oct-7-en- 1-yl)pyrrolidine (***ent*-13**): light yellow syrup; yield for 3 steps: 65%; [α]_D_
^20^ + 35.0 (*c* 6.95 in CH_2_Cl_2_); *ν*_max_/cm^−1^: 3031 (w), 2926 (m), 1701 (vs), 1408 (m), 1095 (vs), 697 (m); ^1^H-NMR (400 MHz, CDCl_3_) *δ* 7.37–7.14 (m, 20H), 5.80–5.73 (m, 1H), 5.22–5.14 (m, 1H), 5.02–4.92 (m, 3H), 4.62–4.52 (m, 1.5H), 4.44–4.29 (m, 5H), 4.18 (d, *J* = 8.4 Hz, 1H), 4.11–4.08 (m, 1H), 3.93–3.72 (m, 2.5H), 3.51–3.46 (m, 1H), 2.18–1.84 (m, 2.5H), 1.81–1.48 (m, 1.5H), 1.41–1.05 (m, 8H); ^13^C-NMR (100 MHz, CDCl_3_) *δ* 154.8, 154.4, 139.2, 139.2, 138.8, 138.5, 138.2, 138.1, 137.9, 137.9, 136.9, 136.8, 128.7, 128.62, 128.59, 128.54, 128.51, 128.33, 128.25, 128.2, 128.1, 127.94, 127.88, 127.82, 127.76, 127.7, 127.6, 114.54, 114.49, 84.6, 83.5, 83.3, 82.1, 73.2, 73.1, 71.3, 71.2, 71.0, 69.0, 68.0, 67.0, 66.9, 65.3, 64.9, 63.2, 62.8, 34.0, 33.9, 31.6, 30.3, 29.5, 29.2, 29.1, 28.99, 28.95, 26.69, 26.65; HRMS(ESI) calcd for C_42_H_49_O_5_NNa^+^ [M + Na]^+^ 670.35029, found 670.34998.

Data for (2*S*,3*R*,4*S*,5*S*)-3,4-bis(benzyloxy)-1-benzyloxycarbonyl-2-((benzyloxy)methyl)-5-(oct-7-en- 1-yl)pyrrolidine (***ent*-3-*epi*-13**): light yellow syrup; yield for 3 steps: 64%; [α]_D_
^20^ + 5.1 (*c* 3.45 in CH_2_Cl_2_); *ν*_max_/cm^−1^: 3030 (w), 2926 (m), 1701 (vs), 1406 (m), 1094 (s), 697 (m); ^1^H-NMR (400 MHz, CDCl_3_) *δ* 7.35–7.19 (m, 20H), 5.82–5.75 (m, 1H), 5.23–4.92 (m, 4H), 4.82–4.51 (m, 5H), 4.40–4.26 (m, 3H), 4.13 (br, 1H), 3.87–3.67 (m, 2.5H), 3.60 (m, 0.5H), 2.08–1.86 (m, 2.5H), 1.64 (s, 0.5H), 1.41–0.92 (m, 9H); ^13^C-NMR (100 MHz, CDCl_3_) *δ* 154.7, 154.6, 139.2, 138.8, 138.5, 138.4, 136.8, 128.6, 128.5, 128.3, 128.2, 128.1, 128.0, 127.82, 127.75, 127.6, 127.5, 127.4, 127.3, 114.5, 81.1, 79.9, 78.1, 77.7, 77.4, 77.1, 73.1, 73.00, 72.8, 72.3, 72.3, 72.0, 70.3, 69.2, 67.0, 66.9, 62.8, 62.40, 58.0, 57.9, 33.9, 33.3, 32.0, 29.4, 29.1, 29.0, 28.9, 26.7, 26.5; HRMS(ESI) calcd for C_42_H_49_O_5_NNa^+^ [M + Na]^+^ 670.35029, found 670.35004.

Data for (2*S*,3*R*,4*R*,5*R*)-3,4-bis(benzyloxy)-1-benzyloxycarbonyl-2-((benzyloxy)methyl)-5-(oct-7- en-1-yl)pyrrolidine (**2-*epi*-13**): light yellow syrup; yield for 3 steps: 71%; [α]_D_
^20^ −4.2 (*c* 1.95 in CH_2_Cl_2_); *ν*_max_/cm^−1^: 3031 (w), 2927 (m), 1702 (vs), 1406 (m), 1095 (s), 697 (m); ^1^H-NMR (400 MHz, CDCl_3_) *δ* 7.35–7.17 (m, 20H), 5.83–5.73 (m, 1H), 5.10 (s, 2H), 5.00–4.91 (m, 2H), 4.61–4.59 (m, 2H), 4.62–4.33 (m, 5H), 4.15–4.12 (m, 1H), 3.92–3.90 (m, 1H), 3.80–3.65 (m, 3H), 2.00–1.98 (m, 2H), 1.78 (m, 1H), 1.62–1.54 (m, 1H), 1.31–1.105 (m, 8H); ^13^C-NMR (100 MHz, CDCl_3_) *δ* 155.8, 139.3, 138.6, 138.2, 138.1, 136.9, 128.6, 128.5, 128.4, 128.0, 127.9, 127.83, 127.79, 127.7, 127.6, 114.3, 84.9, 82.5, 73.4, 73.0, 71.7, 68.2, 67.0, 62.3, 59.1, 33.9, 33.5, 29.5, 29.1, 29.0, 26.0; HRMS(ESI) calcd for C_42_H_49_O_5_NNa^+^ [M + Na]^+^ 670.35029, found 670.34973.

#### 3.3.6. General Procedure for Synthesis of Compounds **12**, **10’-epi-12**, **ent-10’-epi-12**, **ent-12**, **ent-3,10’-di-epi-12**, **ent-3-epi-12**, **2-epi-12**, and **2,10’-di-epi-12**, with **12** as an Example

Grubbs II catalyst (6.8 mg, 0.008 mmol) was added to a solution of **13** (50 mg, 0.077 mmol) and **15** (25.4 mg, 0.116 mmol) in dry CH_2_Cl_2_ and the resulting reaction mixture was heated to reflux for 4 h. Then the solvent was removed under reduced pressure and purification of the residue by flash chromatography on silica gel (petroleum ether/EtOAc = 5/1) afforded **12** (*Z/E* mixture) as a yellow syrup (28 mg, 43% yield). 

Data for (2*R*,3*R*,4*R*,5*R*)-3,4-bis(benzyloxy)-1-benzyloxycarbonyl-5- ((*R*)-13-(benzyloxy)-10-hydroxydec-7-en-1-yl)-2-((benzyloxy)methyl)pyrrolidine (**12**): [α]_D_
^20^ −20.0 (*c* 1.50 in CH_2_Cl_2_); *ν*_max_/cm^−1^: 3079 (w), 2925 (s), 2854 (m), 1699 (s), 1409 (m), 1094 (s), 697 (m); ^1^H-NMR (400 MHz, CDCl_3_) *δ* 7.45–7.13 (m, 25H), 5.49–5.39 (m, 2H), 5.22–5.15 (dd, *J* = 14.8, 12.6 Hz, 1H), 5.05–5.02 (d, *J* = 12.3 Hz, 1H), 4.65–4.57 (m, 1.5H), 4.50–4.32 (m, 6.5H), 4.2–4.23 (dd, *J* = 10.5, 4.0 Hz, 0.5H), 4.18–4.12 (m, 1.5H), 4.07–4.03 (dd, *J* = 8.7, 4.1 Hz, 0.5H), 3.85–3.71 (m, 1.5H), 3.62–3.58 (d, *J* = 3.8 Hz, 1H), 3.52–3.44 (m, 3H), 2.26–1.87 (m, 5H), 1.79–1.40 (m, 5H), 1.38–1.14 (m, 8H); ^13^C-NMR (100 MHz, CDCl_3_) *δ* 153.6, 153.2, 137.5, 137.3, 136.93, 136.88, 136.63, 135.58, 133.32, 133.25, 131.9, 127.5, 127.4, 127.32, 127.28, 127.1, 126.98, 126.95, 126.64, 126.56, 126.5, 125.0, 83.4, 82.3, 82.1, 81.0, 76.3, 76.0, 75.7, 72.0, 71.9, 71.7, 70.1, 70.0, 69.8, 69.5, 69.4, 67.7, 66.8, 65.6, 64.0, 63.6, 61.9, 61.6, 39.8, 34.4, 32.8, 31.6, 31.1, 30.4, 29.1, 28.7, 28.4, 28.2, 28.1, 28.0, 26.4, 25.5, 25.2, 25.0; HRMS(ESI) calcd for C_54_H_65_O_7_NNa^+^ [M + Na]^+^ 862.46532, found 862.46423.

Data for (2*R*,3*R*,4*R*,5*R*)-3,4-bis(benzyloxy)-1-benzyloxycarbonyl-5-((*S*)-13-(benzyloxy)-10- hydroxydec-7-en-1-yl)-2-((benzyloxy)methyl)pyrrolidine (**10’-*epi*-12**): yellow syrup; yield: 43%; [α]_D_
^20^ −20.0 (*c* 1.50 in CH_2_Cl_2_); *ν*_max_/cm^−1^: 3080 (w), 2927 (s), 2855 (m), 1699 (s), 1409 (m), 1095 (s), 697 (m); ^1^H-NMR (400 MHz, CDCl_3_) *δ* 7.33–7.18(m, 25H), 5.54–5.38 (m, 2H), 5.22–5.15(m, 1H), 5.05–5.02 (m, 1H), 4.65–4.57 (m, 1.5H), 4.51–4.32 (m, 6.5H), 4.26–4.22 (m, 0.5H), 4.16–4.11 (m, 1.5H), 4.07–4.03 (m, 0.5H), 3.86–3.71 (m, 2.5H), 3.62–3.58 (m, 1H), 3.53–3.44 (m, 3H), 2.24–1.86 (m, 5H), 1.74–1.61(m, 4H), 1.52–1.38 (m, 1H), 1.36–1.16 (m, 8H); ^13^C-NMR (100 MHz, CDCl_3_) *δ* 154.7, 154.3, 138.61, 138.36, 138.03, 137.98, 137.76, 136.67, 134.30, 134.23, 132.98, 130.02, 128.6, 128.49, 128.46, 128.4, 128.2, 128.10, 128.06, 127.8, 127.74, 127.69, 127.65, 127.6, 126.1, 84.5, 83.4, 83.2, 82.0, 73.1, 73.0, 72.7, 71.2, 71.1, 70.9, 70.6, 70.5, 68.8, 67.9, 66.91, 66.87, 65.1, 64.8, 63.0, 62.7, 40.9, 33.9, 32.7, 32.6, 31.5, 30.2, 29.5, 29.2, 29.1, 27.5, 26.6, 26.3; HRMS(ESI) calcd for C_54_H_65_O_7_NNa^+^ [M + Na]^+^ 862.46532, found 862.46411.

Data for (2S,3*S*,4*S*,5*S*)-3,4-bis(benzyloxy)-1-benzyloxycarbonyl-5-((*R*)-13-(benzyloxy)-10- hydroxydec-7-en-1-yl)-2-((benzyloxy)methyl)pyrrolidine (***ent*-10’-*epi*-12**): yellow syrup; yield: 41%; [α]_D_
^20^ + 28.6 (*c* 1.85 in CH_2_Cl_2_); *ν*_max_/cm^−1^: 3470 (m), 3031 (w), 2925 (s), 1699 (vs), 1409 (m), 1096 (vs), 697 (m); ^1^H-NMR (400 MHz, CDCl_3_) *δ* 7.35–7.17 (m, 25H), 5.54–5.34 (m, 2H), 5.22–5.15 (m, 1H),5.05–5.02 (m, 1H), 4.65–4.56 (m, 1.5H), 4.52–4.32 (m, 6.5H), 4.26–4.22 (dd, *J* = 4.0 Hz, 0.5H), 4.15–4.10 (m, 1.5H), 4.06–4.03 (dd, *J* = 8.7, 4.2 Hz, 0.5H), 3.87–3.72 (m, 2.5H), 3.64–3.56 (m, 1H), 3.52–3.44 (m, *3*H), 2.24–1.91 (m, 5H), 1.78–1.58 (m, 4H), 1.52–1.48 (m, 1H), 1.35–1.11 (m, 8H); ^13^C-NMR (100 MHz, CDCl_3_) *δ* 154.7, 154.3, 138.6, 138.4, 138.0, 138.0, 137.7, 137.6, 136.7, 134.3, 134.2, 129.7, 129.6, 128.6, 128.50, 128.47, 128.4, 128.2, 128.10, 128.07, 127.74, 127.65, 127.6, 127.5, 126.1, 84.5, 83.4, 83.2, 82.0, 73.1, 73.0, 71.2, 71.1, 70.9, 70.6, 70.5, 68.8, 67.8, 66.9, 65.1, 64.8, 63.0, 62.7, 40.9, 33.9, 32.7, 31.5, 30.2, 29.5, 29.2, 29.1, 26.6, 26.3, 26.1; HRMS(ESI) calcd for C_54_H_65_O_7_NNa^+^ [M + Na]^+^ 862.46532, found 862.46407.

Data for (2S,3*S*,4*S*,5*S*)-3,4-bis(benzyloxy)-1-benzyloxycarbonyl-5-((*S*)-13-(benzyloxy)-10- hydroxydec-7-en-1-yl)-2-((benzyloxy)methyl)pyrrolidine (***ent*-12**): yellow syrup; yield: 45%; [α]_D_
^20^ + 27.8 (*c* 1.39 in CH_2_Cl_2_); *ν*_max_/cm^−1^: 3462 (m), 3031 (w), 2925 (s), 1699 (vs), 1409 (m), 1096 (vs), 697 (m); ^1^H-NMR (400 MHz, CDCl_3_) *δ* 7.41–7.16 (m, 25H), 5.54–5.33 (m, 2H), 5.22–5.14 (m, 1H), 5.04–5.01 (m, 1H), 4.65–4.55 (m, 1.5H), 4.51–4.32 (m, 6.5H), 4.27–4.23 (m, 0.5H), 4.15–4.12 (m, 1.5H), 4.06–4.03 (dd, *J* = 8.2, 3.9 Hz, 0.5H), 3.85–3.69 (m, 2.5H), 3.59–3.54(m, 1H), 3.51–3.47 (m, 3H), 2.22–1.90 (m, 5H), 1.77–1.52 (m, 5H), 1.38–1.12 (m, 8H); ^13^C-NMR (100 MHz, CDCl_3_) *δ* 154.7, 154.3, 138.6, 138.3, 138.00, 137.95, 137.7, 136.6, 134.34, 134.28, 128.5, 128.44, 128.41, 128.37, 128.2, 128.1, 127.73, 127.69, 127.65, 127.6, 126.1, 84.5, 83.4, 83.2, 82.03, 77.5, 77.2, 76.8, 73.1, 73.0, 71.2, 71.1, 70.9, 70.54, 70.45, 68.79, 67.84, 66.9, 65.1, 64.7, 62.973.0, 62.6, 40.8, 35.4, 33.9, 32.7, 32.2, 31.5, 30.1, 29.4, 29.3, 29.2, 29.1, 27.5, 26.6, 26.3, 26.0; HRMS(ESI) calcd for C_54_H_65_O_7_NNa^+^ [M + Na]^+^ 862.46532, found 862.46418.

Data for (2*S*,3*R*,4*S*,5*S*)-3,4-bis(benzyloxy)-1-benzyloxycarbonyl-5-((*R*)-13-(benzyloxy)-10- hydroxydec-7-en-1-yl)-2-((benzyloxy)methyl)pyrrolidine (***ent*-3,10’-di-*epi*-12**): yellow syrup; yield: 40%; [α]_D_
^20^ + 5.1 (*c* 1.05 in CH_2_Cl_2_); *ν*_max_/cm^−1^: 3461 (m), 3030 (w), 2926 (s), 1700 (vs), 1408 (m), 1096 (vs), 697 (m); ^1^H-NMR (400 MHz, CDCl_3_) *δ* 7.38–7.17 (m, 25H), 5.53–5.38 (s, 2H), 5.25–5.00 (m, 2H), 4.85–4.45 (m, 7H), 4.39–4.22 (m, 3H), 4.12–4.17 (m, 1H), 3.81–3.74 (m, 2H), 3.71–3.68 (m, 0.5H), 3.65–3.57 (m, 1.5H), 3.52–3.46 (t, *J* = 6.1 Hz, 2H), 2.26–1.88 (m, 5H), 1.79–1.46 (m, 5H), 1.32–1.01 (m, 8H); ^13^C-NMR (100 MHz, CDCl_3_) *δ* 154.7, 154.5, 138.7, 138.4, 136.7, 134.2, 128.5, 128.5, 128.4, 128.2, 128.0, 127.72, 127.65, 127.5, 127.4, 126.30, 126.2, 81.1, 79.9, 78., 77.5, 77.2, 76.9, 73.0, 72.7, 72.5, 72.2, 71.9, 70.9, 70.6, 69.1, 66.9, 62.7, 62.3, 57.9, 40.9, 37.3, 33.9, 33.3, 32.7, 31.9, 29.4, 29.1, 26.6, 26.3; HRMS(ESI) calcd for C_54_H_65_O_7_NNa^+^ [M + Na]^+^ 862.46532, found 862.46412.

Data for (2*S*,3*R*,4*S*,5*S*)-3,4-bis(benzyloxy)-1-benzyloxycarbonyl-5-((*S*)-13-(benzyloxy)-10- hydroxydec-7-en-1-yl)-2-((benzyloxy)methyl)pyrrolidine (***ent*-3-*epi*-12**): yellow syrup; yield: 41%; [α]_D_
^20^+ 5.9 (*c* 3.3 in CH_2_Cl_2_); *ν*_max_/cm^−1^: 3461 (m), 3030 (w), 2926 (s), 1701 (vs), 1408 (m), 1096 (vs), 697 (m); ^1^H-NMR (500 MHz, CDCl_3_) *δ* 7.38–7.17 (m, 25H), 5.53–5.36 (m, 2H), 5.24–5.01 (m, 2H), 4.82–4.75 (m, 1H), 4.71–4.46 (m, 6H), 4.42–4.34 (m, 1H), 4.31–4.23 (m, 2H), 4.15–4.14 (d, *J* = 3.7 Hz, 1H), 3.81–3.77 (m, 2H), 3.70–3.68 (d, *J* = 8.1 Hz, 0.5H), 3.60–3.57 (m, 1.5H), 3.52–3.46 (m, 2H), 2.22–1.89 (m, 5H), 1.78–1.58 (m, 5H), 1.34–1.01 (m, 8H); ^13^C-NMR (125 MHz, CDCl_3_) *δ* 154.7, 154.5, 1390, 138.7, 138.4, 138.3, 136.7, 136.6, 134.24, 134.17, 132.9, 129.6, 128.50, 128.45, 128.4, 128.2, 128.1, 127.99, 127.95, 127.73, 127.65, 127.5, 127.4, 127.3, 127.2, 126.3, 126.1, 81.0, 79.8, 78.0, 77.6, 77.4, 77.2, 76.9, 73.0, 72.9, 72.7, 72.4, 72.2, 71.9, 71.3, 70.9, 70.5, 70.5, 69.1, 66.9, 62.6, 62.3, 57.9, 57.8, 40.8, 35.5, 34.1, 33.9, 33.3, 32.8, 32.7, 32.6, 32.2, 31.8, 29.4, 29.3, 29.1, 29.0, 27.4, 26.6, 26.5, 26.4, 26.3, 26.0; HRMS(ESI) calcd for C_54_H_65_O_7_NNa^+^ [M + Na]^+^ 862.46532, found 862.46437.

Data for (2*S*,3*R*,4*R*,5*R*)-3,4-bis(benzyloxy)-1-benzyloxycarbonyl-5-((*R*)-13-(benzyloxy)-10- hydroxydec-7-en-1-yl)-2-((benzyloxy)methyl)pyrrolidine (**2-*epi*-12**): yellow syrup; yield: 43%; [α]_D_
^20^ −3.7 (*c* 0.88 in CH_2_Cl_2_); *ν*_max_/cm^−1^: 3461 (m), 3030 (w), 2926 (s), 1700 (vs), 1409 (m), 1097 (vs), 697 (m); ^1^H-NMR (400 MHz, CDCl_3_) *δ* 7.34–7.24 (m, 25H), 5.52–5.38 (m, 2H), 5.11 (s, 2H), 4.64–4.58 (m, 2H), 4.55–4.46 (m, 6H), 4.36–4.32 (m, 1H), 4.15–4.12 (dd, *J* = 6.8, 4.6 Hz, 1H), 3.91–3.89 (t, *J* = 4.0 Hz, 1H), 3.85–3.66 (m, 3H), 3.61–3.57 (m, 1H), 3.52–3.46 (t, *J* = 6.1 Hz, 2H), 2.25–1.92 (m, 4H), 1.81–1.60 (m, 4H), 1.52–1.42 (m, 1H), 1.35–1.12 (m, 8H); ^13^C-NMR (100 MHz, CDCl_3_) *δ* 155.6, 138.5, 138.4, 138.2, 138.0, 136.8, 134.4, 132.9, 128.5, 128.48, 128.45, 128.4, 128.3, 128.0, 127.8, 127.8, 127.7, 127.6, 127.5, 126.0, 89.6, 82.5, 80.7, 73.4, 73.0, 73.0, 72.7, 71.6, 71.3, 70.9, 70.6, 70.5, 67.0, 62.3, 59.0, 40.9, 35.5, 34.1, 33.9, 32.7, 29.5, 29.4, 29.1, 27.5, 26.4, 26.3, 26.0, 25.9; HRMS(ESI) calcd for C_54_H_65_O_7_NNa^+^ [M + Na]^+^ 862.46532, found 862.46422.

Data for (2*S*,3*R*,4*R*,5*R*)-3,4-bis(benzyloxy)-1-benzyloxycarbonyl-5-((*S*)-13-(benzyloxy)-10- hydroxydec-7-en-1-yl)-2-((benzyloxy)methyl)pyrrolidine (**2,10’-di-*epi*-12**): yellow syrup; yield: 43%; [α]_D_
^20^ −3.8 (*c* 1.43 in CH_2_Cl_2_); *ν*_max_/cm^−1^: 3470 (m), 3030 (w), 2926 (s), 1700 (vs), 1409 (m), 1097 (vs), 697 (m); ^1^H-NMR (400 MHz, CDCl_3_) *δ* 7.36–7.24 (m, 25H), 5.52–5.34 (m, 2H), 5.13 (s, 2H), 4.65–4.48 (m, 8H), 4.36–4.32 (m, 1H), 4.15–4.12 (dd, *J* = 6.8, 4.7 Hz, 1H), 3.92–3.90 (t, *J* = 3.9 Hz, 1H), 3.83 –3.67 (m, 3H), 3.52–3.45 (m, 2H), 2.26–1.85 (m, 5H), 1.77–1.41 (m, 5H), 1.35–1.13 (m, 8H); ^13^C-NMR (100 MHz, CDCl_3_) *δ* 153.0, 138.51, 138.45, 138.2, 138.1, 136.8, 134.3, 130.0, 128.53, 128.51, 128.47, 128.46, 128.4, 128.0, 1279, 127.79, 127.76, 127.69, 127.66, 127.5, 126.1, 83.9, 79.9, 73.4, 73.1, 73.0, 72.9, 71.7, 71.3, 71.1, 70.9, 70.64, 70.57, 69.4, 68.1, 67.0, 62.3, 59.1, 40.9, 37.4, 34.8, 34.7, 33.9, 32.8, 29.5, 29.4, 29.2, 26.4, 26.3, 25.9, 25.8; HRMS(ESI) calcd for C_54_H_65_O_7_NNa^+^ [M + Na]^+^ 862.46532, found 862.46448.

#### 3.3.7. General Procedure for Synthesis of Compounds **3**, **10’-epi-3**, **ent-10’-epi-3**, **ent-3**, **ent-3**,**10’-di-epi-3**, **ent-3-epi-3**, **2-epi-3**, **2**, and **10’-di-epi-3**, with Broussonetine **M** (**3**) as an Axample

Pd/C (10 wt%) was added to a stirred solution of **9** (43 mg, 0.055 mmol) and 3 N HCl (1 mL) in MeOH (10 mL) under Ar atmosphere and the reaction mixture was stirred under H_2_ atmosphere for 12 h. Then, the catalyst was filtered and the solvent was removed under reduced pressure. Purification of the residue by flash chromatography on silica gel (CHCl_3_/MeOH/NH_3_·H_2_O (2 N) = 90/9/1) afforded (2*R*,3*R*,4*R*,5*R*)-2-hydroxymethyl-3,4-dihydroxyl-5-((10*R*)-10,13-(dihydroxyl)dec- 1-yl)pyrrolidine **(3)** as white solid (14.4 mg, 0.055 mmol, quantative yield). Data for broussonetine M (**3**): [α]_D_
^20^ + 4.0 (*c* 0.7 in CH_3_OH) [lit. [36] [α]_D_
^20^ + 5.9 (*c* 0.3 in CH_3_OH)]; *ν*_max_/cm^−1^: 3104 (vs), 2926 (s), 2857 (w),1402 (vs); ^1^H-NMR (500 MHz, C_5_D_5_N) *δ* 6.41 (br, 7.15–5.54, 4H), 4.92 (t, *J* = 6.7 Hz, 1H), 4.67 (t, *J* = 7.1 Hz, 1H), 4.49–4.42 (m, 2H), 4.31–4.28 (m, 1H), 4.09–4.05 (m, 1H), 3.96 (t, *J* = 6.4 Hz, 2H), 3.92–3.87 (m 1H), 2.34–2.29 (m, 2H), 2.15–2.09 (m, 1H), 2.03–1.97 (m, 1H), 1.90–1.79 (m, 3H), 1.74–1.56 (m, 4H), 1.48–1.45 (m, 1H), 1.33–1.07 (m, 10H); ^13^C-NMR (125 MHz, C_5_D_5_N) *δ* 80.3, 76.2, 70.8, 65.0, 62.7, 62.3, 59.2, 38.3, 35.1, 31.5, 30.2, 29.9, 29.7, 29.5, 29.4, 29.3, 26.6, 26.1; ^1^H-NMR (400 MHz, CD_3_OD) *δ* 4.00–3.97 (t, *J* = 6.1 Hz, 1H), 3.90–3.80 (m, 3H), 3.58–3.48 (m, 4H), 3.34 (m, 1H), 1.98–1.28 (m, 22H); ^13^C-NMR (100 MHz, CD_3_OD) *δ* 80.3, 76.6, 72.3, 65.5, 64.3, 63.1, 59.5, 38.4, 34.7, 32.0, 30.8, 30.7, 30.6, 30.39, 30.35, 29.9, 27.2, 26.8; HRMS(ESI) calcd for C_18_H_38_O_5_N^+^ [M + H]^+^ 348.27445, found 348.27432.

Data for (2*R*,3*R*,4*R*,5*R*)-2-hydroxymethyl-3,4-dihydroxyl-5-((10*S*)-10,13-(dihydroxyl)dec-1-yl) pyrrolidine (**10’-*epi*-3**): white solid, quantative yield; [α]_D_
^20^ + 4.0 (*c* 0.4 in CH_3_OH); *ν*_max_/cm^−1^: 3104 (vs), 2927 (s), 2861 (w), 1402 (vs); ^1^H-NMR (400 MHz, CD_3_OD) *δ* 3.99–3.96 (t, *J* = 5.7 Hz, 1H), 3.85 (m, 3H), 3.62–3.51 (m, 3H), 3.49–3.45 (m, 1H), 3.37–3.35 (m, 1H), 1.96–1.84 (m, 1H), 1.80–1.64 (m, 2H), 1.63–1.28 (m, 19H); ^13^C-NMR (100 MHz, CD_3_OD) *δ* 80.3, 76.6, 72.3, 65.5, 64.3, 63.2, 59.6, 38.4, 34.8, 32.2, 30.8, 30.7, 30.6, 30.4, 29.9, 27.2, 26.8; HRMS(ESI) calcd for C_18_H_38_O_5_N^+^ [M + H]^+^ 348.27445, found 348.27411.

Data for (2*S*,3*S*,4*S*,5*S*)-2-hydroxymethyl-3,4-dihydroxyl-5-((10*R*)-10,13-(dihydroxyl)dec-1-yl) pyrrolidine (***ent*-10’-*epi*-3**): white solid, quantative yield; [α]_D_
^20^ −16.8 (*c* 1.5 in CH_3_OH); *ν*_max_/cm^−1^: 3134 (vs), 2928 (s), 2854 (w), 1402 (vs); ^1^H-NMR (400 MHz, CD_3_OD) *δ* 4.01–3.98 (t, *J* = 6.2 Hz, 1H), 3.91–3.80 (m, 3H), 3.59–3.47 (m, 4H), 3.36–3.33 (m, 1H), 1.95–1.30 (m, 22H); ^13^C-NMR (100 MHz, CD_3_OD) *δ* 80.3, 76.6, 72.3, 65.5, 64.3, 63.1, 59.5, 38.4, 34.8, 32.0, 30.8, 30.7, 30.6, 30.39, 30.35, 29.9, 27.2, 26.8; HRMS(ESI) calcd for C_18_H_38_O_5_N^+^ [M + H]^+^ 348.27445, found 348.27430.

Data for (2*S*,3*S*,4*S*,5*S*)-2-hydroxymethyl-3,4-dihydroxyl-5-((10*S*)-10,13-(dihydroxyl)dec-1-yl) pyrrolidine (***ent*-3**): white solid, quantative yield; [α]_D_
^20^ −20.4 (*c* 1.55 in CH_3_OH); *ν*_max_/cm^−1^: 3134 (vs), 2928 (s), 2854 (w), 1403 (s); ^1^H-NMR (400 MHz, CD_3_OD) *δ* 4.01–3.98 (t, *J* = 6.1 Hz, 1H), 3.90–3.80 (m, 3H), 3.58–3.45 (m, 4H), 3.36–3.33 (m, 1H), 1.94–1.29 (m, 22H); ^13^C-NMR (100 MHz, CD_3_OD) *δ* 80.3, 76.6, 72.3, 65.5, 64.3, 63.1, 59.5, 38.4, 34.8, 32.0, 30.8, 30.7, 30.6, 30.39, 30.35, 29.9, 27.2, 26.8; HRMS(ESI) calcd for C_18_H_38_O_5_N^+^ [M + H]^+^ 348.27445, found 348.27416.

Data for (2*S*,3*R*,4*S*,5*S*)-2-hydroxymethyl-3,4-dihydroxyl-5-((10*R*)-10,13-(dihydroxyl)dec-1-yl) pyrrolidine (***ent*-3,10’-di-*epi*-3**): white solid, quantative yield; [α]_D_
^20^ −19.4 (*c* 0.5 in CH_3_OH); *ν*_max_/cm^−1^: 3124 (vs), 2926 (s), 2853 (w), 1402 (s); ^1^H-NMR (500 MHz, CD_3_OD) *δ* 4.16 (s, 1H), 3.97–3.92 (m, 2H), 3.90–3.85 (m, 1H), 3.70–3.68 (dd, *J* = 8.0, 3.9 Hz, 1H), 3.59–3.54 (m, 3H), 3.46–3.41 (td, *J* = 9.1, 5.2 Hz, 1H), 1.89–1.29 (m, 22H); ^13^C-NMR (125 MHz, CD_3_OD) *δ* 77.4, 72.3, 71.7, 63.5, 63.1, 62.0, 59.4, 38.5, 34.8, 32.0, 30.8, 30.7, 30.6, 30.46, 30.38, 29.9, 27.6, 26.8; HRMS(ESI) calcd for C_18_H_38_O_5_N^+^ [M + H]^+^ 348.27445, found 348.27424.

Data for (2*S*,3*R*,4*S*,5*S*)-2-hydroxymethyl-3,4-dihydroxyl-5-((10*S*)-10,13-(dihydroxyl)dec-1-yl) pyrrolidine (***ent*-3-*epi*-3**): white solid, quantative yield; [α]_D_
^20^ −33.4 (*c* 0.65 in CH_3_OH); *ν*_max_/cm^−1^: 3364 (vs), 2925 (vs), 2852 (m), 1402 (s); ^1^H-NMR (400 MHz, CD_3_OD) *δ* 4.16 (t, *J* = 3.4 Hz, 1H), 3.97–3.85 (m, 3H), 3.68–3.63 (m, 1H), 3.56 (t, *J* = 6.4 Hz, 2H), 3.53 (m, 1H), 3.42 (td, *J* = 9.2, 5.2 Hz, 1H), 1.90–1.28 (m, 22H); ^13^C-NMR (100 MHz, CD_3_OD) *δ* 77.4, 72.3, 71.7, 63.5, 63.1, 62.0, 59.4, 38.5, 34.8, 32.08, 30.8, 30.7, 30.6, 30.5, 30.4, 29.9, 27.6, 26.8; HRMS(ESI) calcd for C_18_H_38_O_5_N^+^ [M + H]^+^ 348.27445, found 348.27427.

Data for (2*S*,3*R*,4*R*,5*R*)-2-hydroxymethyl-3,4-dihydroxyl-5-((10*R*)-10,13-(dihydroxyl)dec-1-yl) pyrrolidine (**2-*epi*-3**): white solid, quantative yield; [α]_D_
^20^ + 3.1 (*c* 0.45 in CH_3_OH); *ν*_max_/cm^−1^: 3124 (s), 2923 (s), 2852 (w), 1402 (s); ^1^H-NMR (500 MHz, CD_3_OD) *δ* 4.07 (s, 1H), 3.98–3.89 (m, 3H), 3.72–3.68 (m, 1H), 3.56 (t, *J* = 6.3 Hz, 2H), 3.54–3.52 (m, 1H), 3.30–3.26 (m, 1H), 1.88–1.29 (m, 22H); ^13^C-NMR (125 MHz, CD_3_OD) *δ* 80.8, 76.5, 72.3, 68.7, 65.6, 63.1, 58.6, 38.5, 34.8, 33.1, 30.8, 30.7, 30.6, 30.5, 30.3, 29.9, 27.8, 26.8; HRMS(ESI) calcd for C_18_H_38_O_5_N^+^ [M + H]^+^ 348.27445, found 348.27439.

Data for (2*S*,3*R*,4*R*,5*R*)-2-hydroxymethyl-3,4-dihydroxyl-5-((10*S*)-10,13-(dihydroxyl)dec-1-yl) pyrrolidine (**2,10’-di-*epi*-3**): white solid, quantative yield; [α]_D_
^20^ + 0.12 (*c* 1.2 in CH_3_OH); *ν*_max_/cm^−1^: 3127 (s), 2923 (s), 2853 (m), 1403 (s); ^1^H-NMR (400 MHz, CD_3_OD) *δ* 4.08 (d, *J* = 2.0 Hz, 1H), 3.99–3.88 (m, 3H), 3.71 (m, 1H), 3.60–3.50 (m, 3H), 3.31–3.27 (m, 1H), 1.93–1.76 (m, 2H), 1.71–1.28 (m, 20H); ^13^C-NMR (100 MHz, CD_3_OD) *δ* 80.8, 76.5, 72.3, 68.7, 65.5, 63.1, 58.6, 38.4, 34.8, 33.1, 30.8, 30.7, 30.6, 30.4, 30.3, 29.9, 27.8, 26.8; HRMS(ESI) calcd for C_18_H_38_O_5_N^+^ [M + H]^+^ 348.27445, found 348.27429.

## 4. Conclusions

In summary, a general and versatile synthetic strategy has been developed for the synthesis of broussonetine M (**3**), *ent*-broussonetine M (***ent*-3**), and six other stereoisomers with d-*arabino*-nitrone (14), l-*arabino*-nitrone (***ent*-14**), l-*lyxo*-nitrone (***ent*-3-*epi*-14**), and l-*xylo*-nitrone (**2-*epi*-14**) as the starting materials in 26%–31% total yield for five linear steps. Glycosidase inhibition assays on a range of enzymes showed that natural product broussonetine M (**3**) and **10’-*epi*-3** showed potent inhibition of β-glucosidase from bovine liver, while ***ent*-3** and ***ent*-10’-*epi*-3** were potent and selective inhibitors of rice α-glucosidase and rat intestinal maltase. It was also found that the configuration at C-3 was essential for α-glucosidase inhibition. This work has further explored the spectrum of glycosidase inhibition by DAB- and LAB-related iminosugars and will be helpful for the future design of potent and selective glycosidase inhibitors.

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
