# Peer review of "Synthesis and Glycosidase Inhibition of Broussonetine M and Its Analogues"

_molecules, 2019, doi:10.3390/molecules24203712_

Round 1
Reviewer 1 Report
The authorsdescribe the chemical synthesis of a small library of iminosugars structurally-based on broussonetine, and inhibition of some glycosidases.
The chemical approach proposed herein is clearly presented. The synthesis, starting from pentose-nitrones as key precursors, are efficient and NMR spectra given in the SI established structures and purity of the compounds.
The resulting compounds were further tested as inhibitors of alpha- and beta-gluco-, galacto- and mannosidases, but also of fucosidase, trehalase, amyloglucosidase, rhamnosidase and glucuronidase. The inhibition results are moderate, and the synthetic anaogus of broussonetine unfortunately less active than other iminosugars. Nevertheless, it is interesting to note the role of the hydroxyl group present on the lipophilic chain underlined by the authors. This point deserves to be deepened in terms of structure-activity relationships.
Experimental part: nothing to say, except maybe the wish of the referee to see complete assignment of NMR data, considering the quality of the spectra given in the SI.
In conclusion, this paper is suitable for publication in Molecules after minor complements.
Author Response
please see the attached file for reply to reviewer 1's questions.

Reviewer 2 Report
Please see the attached file.

Author Response
please see the attached file for reply to reviewer 2's questions.

Reviewer 3 Report
In this manuscript, the authors synthesized several compounds and found that broussonetine M (3) and 10’-epi-3 were potent inhibitors of β-glucosidase and β-galactosidase while their enantiomers, ent-3 and ent-10’-epi-3, were selective and potent inhibitors of rice α-glucosidase and rat intestinal maltase. The methods and results are sound, however, several major points should be addressed.
Major points:
1. The authors evaluated the effects of the compounds on inhibiting various enzymes, and found that several compounds were selective for inhibition of different enzymes. The ATP binding pocket is always used to design enzyme inhibitors. I wonder whether these compounds are ATP competitive inhibitors. It’s recommended to perform a ATP competition assay to investigate this.
2. The authors found that M(3) and 10’-epi-3 were selective for inhibition of β-glucosidase and β-galactosidase while ent-3 and ent-10’-epi-3, were selective and potent inhibitors of rice α-glucosidase and rat intestinal maltase. A structure-activity relationship needs to be discussed in more detail.
3. What are the effects of D- and L- configurations of the enantiomers on inhibition activity? The authors need to discuss this in more detail.
4. β-Galactosidase (β-gal) is an important biomarker for cancers, I’m interested in the effects of M (3) and 10’-epi-3 in cells. It’s recommended to evaluate the cytotoxicity of M3 in cancer cells and normal cells. Some cancer cells high express β-galactosidase (β-gal) which can be considered as the cell model. Please see:
1. Analytical Chemistry 89.21 (2017): 11679-11684.
2. Aging Cell 5.2 (2006): 187-195.
3. Nature Communications 6 (2015): 6463.
4. Human Gene Therapy 9.12 (1998): 1769-1774.
Overall, I recommend publication of this manuscript after major revision.
Author Response
please see the attached file for reply to reviewer 3's questions.

Round 2
Reviewer 3 Report
The authors have satisfactorily responded to all the questions and made the necessary changes to the manuscript. I have no further questions and suggest the acceptance of the revised manuscript.